# The Impacts of Clinical Pharmacists’ Interventions on Clinical Significance and Cost Avoidance in a Tertiary Care University Hospital in Oman: A Retrospective Analysis

**DOI:** 10.3390/pharmacy10050127

**Published:** 2022-10-03

**Authors:** Juhaina Salim Al-Maqbali, Aqila Taqi, Samyia Al-Ajmi, Buthaina Al-Hamadani, Farhat Al-Hamadani, Fatima Bahram, Kifah Al-Balushi, Sarah Gamal, Esra Al-Lawati, Bushra Al Siyabi, Ekram Al Siyabi, Nashwa Al-Sharji, Ibrahim Al-Zakwani

**Affiliations:** 1Department of Pharmacology and Clinical Pharmacy, College of Medicine & Health Sciences, Sultan Qaboos University, Muscat PC 123, Oman; 2Department of Pharmacy, Sultan Qaboos University Hospital, Muscat PC 123, Oman

**Keywords:** clinical pharmacists, clinical significance, cost savings, health care costs, interventions, Oman

## Abstract

Objectives: Pharmaceutical interventions are implicit components of the enhanced role that clinical pharmacists provide in clinical settings. We aimed to study the clinical significance and analyze the presumed cost avoidance achieved by clinical pharmacists’ interventions. Methods: A retrospective study of documented clinical pharmacists’ interventions at a tertiary care hospital in Oman was conducted between January and March 2022. The interventions were electronically recorded in the patients’ medical records as routine practice by clinical pharmacists. Data on clinical outcomes were extracted and analyzed. Cost implications were cross checked by another clinical pharmacist, and then, cost avoidance was calculated using the Rx Medi-Trend system values. Results: A total of 2032 interventions were analyzed, and 97% of them were accepted by the treating physicians. Around 30% of the accepted interventions were for antimicrobials, and the most common type was dosage adjustment (30%). Treatment efficacy was enhanced in 60% and toxicity was avoided in 22% of the interventions. The presumed cost avoided during the study period was USD 110,000 with a projected annual cost avoidance of approximately USD 440,000. Conclusion: There was an overall positive clinical and financial impact of clinical pharmacists’ interventions. Most interventions have prevented moderate or major harm with a high physician acceptance rate. Optimal documentation of the interventions is crucial for emphasizing clinical pharmacists’ value in multi-specialty hospitals.

## 1. Introduction

Clinical pharmacists in multidisciplinary care teams play an integral part in improving the quality of medicines’ use processes and in reducing medication errors and adverse drug events (ADEs), as well as enhancing patient health outcomes through effective interaction with both patients and other health professionals [1,2]. Pharmaceutical interventions are implicit components of the enhanced role that clinical pharmacists play in clinical settings and are defined as the actions that identify and prevent medication-related problems and optimize patient medication therapy [3]. Ample evidence supports the value of clinical pharmacists’ interventions in optimizing medication adherence and clinical outcomes including reduced length of hospital stay (LOS) as well as hospital readmissions [2,4,5,6,7,8].

Worldwide, health care resources and its burden on government and individuals have drawn great attention among health care workers. Studies showed that polypharmacy is highly associated with health harm and hence with increment in the overall medical cost [9,10]. Medication errors are also associated with increasing cost of hospitalization measured by increasing LOS [11], while the economic implications of ADEs and inappropriate drug prescribing are the main health cost drivers [12].

At Sultan Qaboos university hospital (SQUH), Muscat, Oman, clinical pharmacy services have been provided to most clinical/medical specialties since the hospital was opened in 1986. Clinical pharmacist interventions are carried out to optimize pharmaceutical care from patients’ admission until discharge; however, the clinical and economic impact of clinical pharmacists’ interventions at SQUH have not been evaluated. Thus, we aimed to evaluate the type and clinical significance of pharmacist interventions on inpatient prescriptions generated within various hospital specialties. We further analyzed the presumed cost avoidance achieved.

## 2. Methods

### 2.1. Study Design and Setting

This was a retrospective analysis of clinical pharmacist interventions at SQUH, a tertiary care university hospital in Oman, between January and March 2022. At SQUH, clinical pharmacists work with specialist teams from both adult and pediatric units, and their interventions on pharmaceutical issues are documented on a specific form developed at SQUH that was adopted from the American College of Clinical Pharmacy Practice-Based Research Network Medication Error Detection [13] and is incorporated in the hospital’s electronic patient record (EPR).

We retrieved interventions that were originally documented and classified into several measures by 14 clinical pharmacists over the study period. The captured information included the following: the admitting specialty, the prescriber’s’ designation, the types and outcomes of the interventions, the clinical significance, the grading of the clinical significance, and the cost implication associated with each intervention. Forms with incomplete or missing information were excluded. The following study measures were analyzed, although only cost implication was cross checked by another clinical pharmacist.

### 2.2. Study Measures

The following study measures were obtained for each intervention.

The outcome: stratified into accepted, accepted with changes, rejected, or unknown.The clinical significance: stratified into efficacy improved, efficacy reduced, toxicity reduced, unnecessary exposure avoided, or not known.The grading of the clinical significance: stratified into death or major, moderate, minor, or suboptimal standard of care/practice.

A death intervention was considered when an intervention was judged to have prevented a death or a major permanent injury or organ damage. A major intervention may have prevented temporary injury, harm, an increased LOS, readmission, or morbidity and required a major correctional treatment. Moderate and minor interventions were those that had prevented moderate or minor injuries or harm that require temporary simple treatment. Suboptimal standard of care/practice was any intervention that would not prevent any harm but led to a better care practice [14,15].

### 2.3. Presumed Cost Avoidance Calculation

Cost avoidance was calculated from a model developed by [16]. Cost avoidance values are the best available method for estimating cost avoided by clinical pharmacist interventions. These are calculated using the Rx Medi-Trend system values that were assigned to each intervention type based on literature reporting the frequency and average cost of an ADE if no intervention by clinical pharmacist had been carried out in addition to a probability factor of a hospital stay [16]. Presumed cost avoidance refers to an intervention that reduces or eliminates additional expenditure that otherwise may have been incurred in the absence of the intervention [17]. To enable adoption of this model, interventions included for cost impact analysis were re-categorized to match the classification described by [16] (Table 1). The re-categorization was performed independently by a panel of three senior clinical pharmacists and agreed upon by the clinical pharmacists who originally documented the interventions (Table 1). Given that the calculation of the direct cost reduction of an intervention was not practically possible, published cost avoidance values were the most feasible available method for estimating costs avoided by clinical pharmacists’ interventions in our setting.

Cost significance was classified into the following four types:Direct drug/investigation cost reduction: include interventions that are directly associated with absolute cost reduction such as discontinuation of unnecessary medicines, switching to less expensive agents, or altering the route of administration.In-direct drug/investigation cost reduction: include interventions that are associated with in-direct cost avoidance or reduction in LOS.Resource saving: include interventions that are associated with saving nursing, doctor, or pharmacist times or interventions that improve patients’ compliance with medications.Not applicable: include interventions that are not impacting the cost but rather are associated with better care/practice.

### 2.4. Inclusion and Exclusion Criteria for Analyses of Clinical and Cost Impacts

Interventions with ‘accepted’ and ‘accepted with changes’ outcomes were included for clinical impact analysis (measured by clinical significance and grading of clinical significance). For cost impact analyses, the direct drug/investigation cost reduction, indirect drug/investigation cost reduction, and resource saving were included. The presumed cost avoidance of accepted or accepted with changes interventions were included, while interventions with (not applicable) cost significance were excluded (Figure 1).

### 2.5. Statistical Analysis

Descriptive statistics were used to describe the data. For categorical variables, frequencies and percentages were reported. For continuous variables, mean and standard deviation were used to summarize the data. Statistical analyses were conducted using STATA version 16.1 (STATA Corporation, College Station, TX, USA).

### 2.6. Ethics Approval

The study was approved by the Medical and Research Ethics Committee at the College of Medicine and Health Sciences, Sultan Qaboos University, Muscat, Oman (MREC #2657; SQU-EC/648/2021; dated: 14 December 2021).

## 3. Results

A total of 2032 clinical pharmacists’ interventions were documented for 959 inpatients during the 3 months of the study period. The mean age was 46.8 ± 24.8 years and 50.4% of the interventions were performed in male patients. A third of the interventions (32.7%) were in patients admitted under the acute medical specialties, followed by intensive care unit (ICU) (16.9%), and hematology specialties (11.7%). Antimicrobials were the most frequently involved drug class with 30% of the total interventions followed by 18.3% interventions involving cardiovascular drugs and 10.6% involving drugs for nutrition and metabolic disorders (Table 2). Adjusting drug doses was the most common type of clinical pharmacists’ interventions (25.3%), followed by recommending the addition of a medication in (11.4%) while optimizing frequency of the regimen occurred in 10.7% of the total interventions (Figure 2). The top three intervention types accounted for 47.5% of total interventions. The majority of clinical pharmacists’ interventions were accepted (82.7%) or accepted with changes (14.2%), that is representing 96.9% of total interventions; however, 2.1% of clinical pharmacists’ recommendations were rejected, and the outcome was not known in 1.0% of the interventions.

For the clinical impact analyses, interventions with an accepted outcome (1969/2032) were analyzed, out of which the efficacy of the prescribed medication was improved in 60.1% (1183/1969), and toxicity was reduced in 22.1% (435/1969) of the interventions (Figure 3). Major harm was avoided in 13.8% and moderate harm was avoided in 58.2% of the accepted interventions (Figure 4). More than half of the accepted interventions resulted in an indirect cost reduction (58.0%), while 29.2% of the accepted interventions resulted in a direct cost reduction, and 5.9% of the accepted interventions resulted in resource saving (Figure 5).

Out of the 1969 accepted interventions, 1824 were analyzed for cost impact (92.6%). The total calculated presumed cost avoidance was USD 109,732.73 during the study period (3 months), with a projected potential annual cost avoidance estimated to be USD 438,931. The potential annual cost avoidance for interventions on dose adjustments was USD 166,150.2, followed by interventions on addition of medications is USD 89,943.92 (Table 3).

## 4. Discussion

This study described the types and clinical outcomes of pharmacists’ interventions and the associated cost avoidance over the study period. To our knowledge, this study is the first that has attempted to describe the cost avoidance achieved by a broad spectrum of clinical pharmacist interventions, including diverse medication classes at a multispecialty teaching hospital in Oman. More than 30% of the interventions were documented for patients admitted under acute medicine specialties including cardiology, rheumatology, neurology and endocrinology. Approximately one third of the interventions involved a single drug class namely antimicrobials, and dose adjustment was the most common type of interventions. The clinical significance was rated as major in 13.8% and as moderate in 58.2% of the interventions with the total presumed cost avoidance of nearly USD 110,000 during the study period.

In our study, adjusting doses was the most common intervention type, which was consistent with findings reported previously in general and specialized settings. Interventions on dose adjustment to therapeutic dose were found to be among the most common intervention categories in a study reporting clinical pharmacists’ intervention in a teaching hospital [18] and in specific clinical settings such as ICUs [19]. Similarly, adjustment of dosing regimen was the most common type of clinical pharmacist intervention carried out on antimicrobials’ use in a teaching hospital in Oman (42%) [14]. Pharmacists are a useful resource for providing accurate dosing information, particularly for patients with renal or hepatic impairment where doses are often modified to suit patients’ clinical conditions and laboratory parameters.

Recommending the addition of a drug was the second most common type of interventions in the present study and was in line with results of a previous study [17]. Identification of omission of patients’ regular medications was the most prevalent type of interventions accounting for 65.9% (n = 1820), of total interventions in a French teaching hospital site [17]. These findings highlight the frequency of omission of prescription drugs for hospitalized patients, and hence, emphasize the importance of medication reconciliation at transitions of care including admission, discharge and transfer to a different level.

In contrast to findings from other studies where analgesics [20] and proton pump inhibitors (PPIs) [21] were the most common medication type involved in pharmacists’ interventions, our study found that approximately one-third of the interventions involved antimicrobials. This is similar to a previous study in Oman in which interventions on antimicrobials comprised 20% of total interventions [22]. In fact, it is likely that antimicrobials have been consistently the most commonly involved class at SQUH over the past years, as 26% of total interventions were on antimicrobials in 2018 as well [14]. Analgesics (e.g., paracetamol) were commonly duplicated as regular as well as “when necessary” orders, resulting in higher than the licensed daily doses [20], while interventions on PPIs were largely on suggestions to change to lower cost equivalents or for switching from intravenous to oral administration [17]. The top three antimicrobials requiring interventions were vancomycin recorded in 117, meropenem in 60, and co-amoxiclav in 34 interventions, representing 19.0%, 9.8%, and 5.6% of interventions, respectively. It is not surprising that interventions with vancomycin were the most frequent, given that it requires therapeutic drug monitoring (TDM) and subsequent dose adjustments. TDM is one of the main activities of the clinical pharmacists. This finding highlights a strategic direction that hospital policy makers must consider by implementing antimicrobial stewardship programs led by pharmacists. Effective antimicrobial stewardship programs have proven advantages in curbing the emergence of antimicrobial resistance and controlling hospital budget [4].

In the present study, the overall clinical pharmacist acceptance rate was almost 97.0%, which was higher than the 71.0% rate reported in 2013 at White County Medical Center community hospital in the US [19]. Unlike the practice at SQUH where clinical pharmacists attend rounds and clinical meetings, pharmacists at the community hospital communicated the majority of recommendations via designated form placed in patient charts [19]. Several studies reported pharmacists’ participation in hospital medical rounds as one of the factors that helped increase acceptance rate [23]. The acceptance rate reported in the present study was in line with that reported by studies on the pharmacists’ interventions in specialized settings which ranged between 80% and 95% in infectious diseases settings [3,14].

Clinical pharmacists’ interventions were judged to have enhanced the treatment efficacy in the majority of interventions (60.1%) and unnecessary exposure to drugs was avoided in 9.6%. Unlike a previous study’s findings [3], reporting the proportion of interventions with major and moderate clinical significance to be 6% and 34%, the present study found that 13% and 58% of the interventions were of moderate and major clinical significance, respectively. The difference may be attributed to the differences in study settings between a multispecialty hospital and an infectious diseases ward, hence, the range of medications requiring interventions included a diverse range of drug classes compared to primarily antimicrobials.

Around 30% of the interventions involved direct cost reduction in the medication, while 58.0% involved indirect cost avoidance resulting in an estimated annual cost avoidance of nearly 440,000 USD. The average presumed annual cost avoided per pharmacist works out to be around 29,260 USD. Nonetheless, this estimation is still low as the model used for cost avoidance in the present study was adopted from studies that may have included a narrower range of medications used in specialist hospitals while the cost avoidance related to a wider range of highly specialized medications such as antimicrobials, anticoagulants, and biologics is expected to be of a higher impact. Although research has proven the favorable impact of clinical pharmacists’ interventions on hospital budgets, it remains difficult to elucidate which interventions were the most cost effective [24,25]. Cost avoidance measures such as preventing ADEs and subsequent health care utilization, were suggested to have the greatest cost-benefit ratio compared to cost-saving interventions [26]. Findings of this study may lay the groundwork for future economic models in specialized clinical pharmacy areas such as intensive care or surgical specialties.

The main limitations of this study were including interventions documented during a short period of 3 months. A longer follow-up including more hospitals throughout the Sultanate of Oman would give more reliable and generalizable results. Interventions on allergies and adverse drug reactions (ADR) are recorded in different forms in our hospital and were not included in this study, which we believe to be the main contributing factor to our underestimated cost avoidance. Furthermore, the shortage of the number of clinical pharmacists may have impaired the ability to document all the interventions performed on a daily basis, resulting in the underestimation of the number of captured interventions. In addition, there could be some variations among different clinical pharmacists in the quality of documentation, especially on variables involving grading. Moreover, the intervention outcome, clinical significance, and grading did not go through any peer review process.

## 5. Conclusions

The present study showed an overall positive clinical and cost impact of clinical pharmacist interventions at SQUH in Oman. This was a result of the diverse range of intervention types and the high physician acceptance rate. The majority of interventions prevented moderate or major harm and resulted in a projected annual cost saving of approximately 440,000 USD. The optimal documentation of interventions by clinical pharmacists is crucial as it is the only tool for measuring the value of the clinical pharmacist in the multidisciplinary team of any tertiary hospital.

## Figures and Tables

**Figure 1 pharmacy-10-00127-f001:**
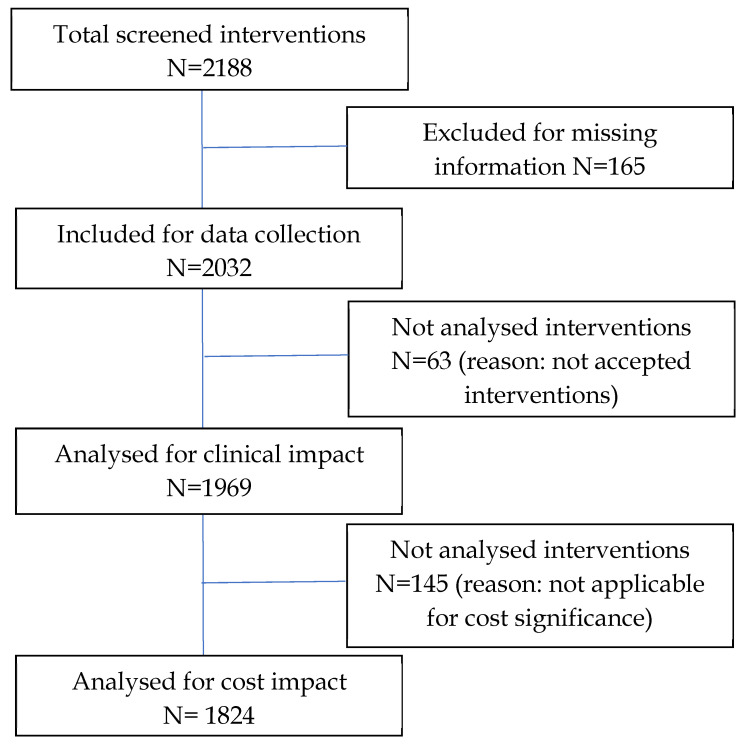
Inclusion and exclusion criteria of the clinical and cost impact analysis.

**Figure 2 pharmacy-10-00127-f002:**
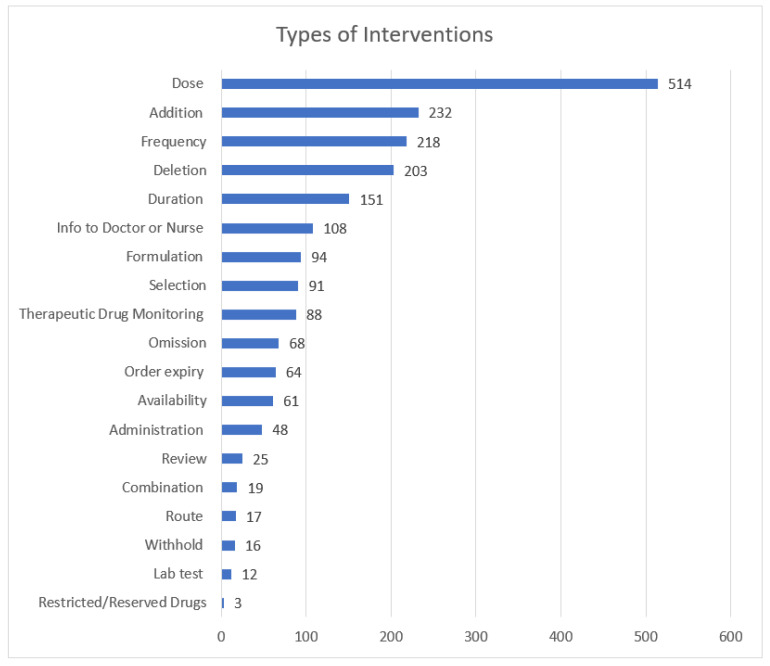
Types of interventions according to the consensus opinion of senior pharmacists at SQUH (N = 2032).

**Figure 3 pharmacy-10-00127-f003:**
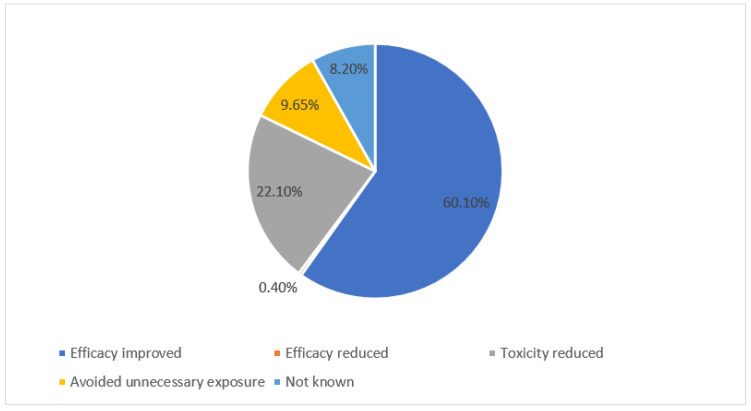
Clinical significance of the total accepted pharmacists’ interventions (N = 1969).

**Figure 4 pharmacy-10-00127-f004:**
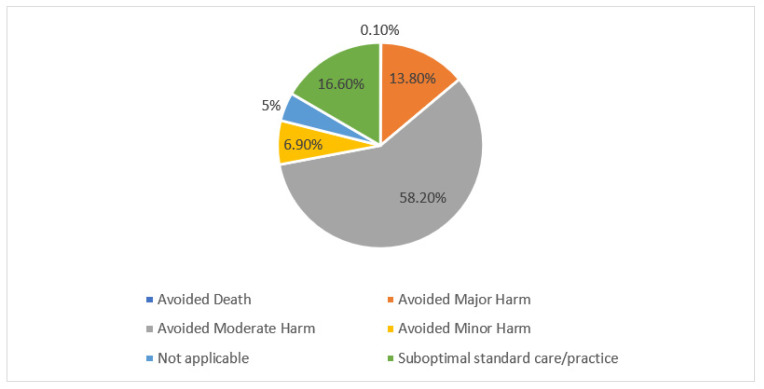
Grading of clinical significance of the total accepted pharmacists’ interventions (N = 1969).

**Figure 5 pharmacy-10-00127-f005:**
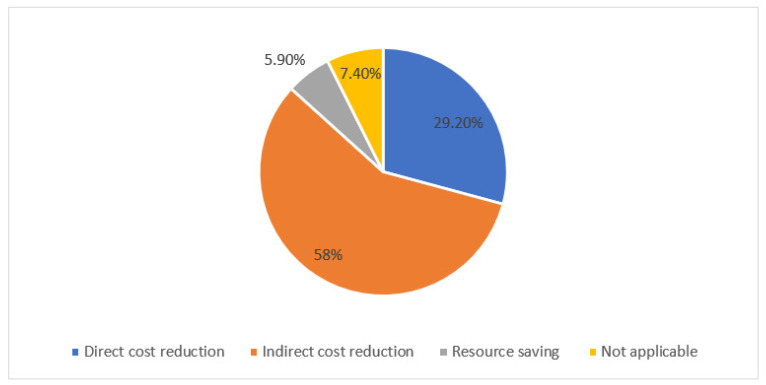
Proportion of clinical pharmacist interventions with cost significance of the total accepted pharmacists’ interventions (N = 1969).

**Table 1 pharmacy-10-00127-t001:** Rearrangement of the classes of the interventions according to SQUH to match the classification of interventions as described by [16].

Intervention Type according to the Paper (Campbell)	Intervention Type according to (SQUH)	Cost per Intervention of Respective Type ($)
Addition	Addition, order expiry, or extended duration	62.81
Change medication	Selection or availability: A change of formulary item to another formulary item due to non-availability.	61.75
Discontinuation of a medication	Deletion (includes double order) or reduced duration	50.39
Dosage form change	Formulation, IV-oral, or route	54.33
Dose adjustment	Dose or frequency	61.72
Drug information	Information to doctor or nurse, administration (includes timing)	24.64
Medication reconciliation	Withhold or re-start (includes omission)	27.58
Monitoring laboratory order	Lab request	77.92
Non-formulary consultation	Restricted/Revered (specialty consultation regarding a non-formulary/reserved item approval for use)	47.86
Non-formulary to formulary conversion	Restricted/Revered (a change of non-formulary item to formulary item without a specialty consultation) or availability (A change of non-formulary item to formulary/or non- formulary due to non-availability)	36.73
Pharmacokinetic monitoring—level adjustment	TDM request, TDM follow up, or TDM dose	77.45
Prevention of adverse drug event	Combination (includes contraindication, interaction, combination of meds, or therapeutic duplication)	470.99
Prompted medical follow-up	Review (includes referral)	56.58

ADR; Adverse Drug Reaction, TDM; Therapeutic Drug Monitoring.

**Table 2 pharmacy-10-00127-t002:** Most common drug classes involved in clinical pharmacists’ interventions (N = 2032).

Class of the Involved Drug	Number of Interventions	(%)
Antimicrobial	610	30.1
Cardiovascular system (including anticoagulants)	371	18.3
Nutrition & Metabolic disorders	215	10.6
Endocrine system	210	10.3
Gastrointestinal system	149	7.3
Nervous system	123	6.0
Respiratory system	77	3.7
Analgesia (including opioids)	66	3.2
Blood disorders and Immunoglobulins	60	2.9
Cytotoxic drugs/Immunosuppressants	47	2.3
Genito-urinary system	20	1.0
Skin preparation	17	0.8
Vaccine	16	0.8
Musculoskeletal system	12	0.6
Ear, eye, nose & oropharynx	19	0.9
Anesthesia	3	0.1

**Table 3 pharmacy-10-00127-t003:** Presumed cost avoidance of clinical pharmacist interventions (N = 1824).

Type of Intervention according to Campbell *	Frequency over the Study Period (3 Months)	Total Presumed Cost Avoidance during the Study Period ($) (3 Months)	Potential Annual Cost Avoidance ($)
Addition	358	22,485.98	89,943.92
Change medication	106	6545.5	26,182
Discontinuation of a medication	263	13,252.57	53,010.28
Dosage form change	81	4400.73	17,602.92
Dose adjustment	673	41,537.56	16,6150.2
Drug information	127	3129.28	12,517.12
Medication reconciliation	81	2233.98	8935.92
Monitoring laboratory order	11	857.12	3428.48
Non-formulary consultation	3	143.58	574.32
Non-formulary to formulary conversion	6	220.38	881.52
Pharmacokinetic monitoring—level adjustment	83	6428.35	25,713.4
Prevention of adverse drug event	16	7535.84	30,143.36
Prompted medical follow-up	17	961.86	3847.44
Total	1824	109,732.73	438,931

* [16].

## Data Availability

Not applicable.

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
