# Peer review of "The Impacts of Clinical Pharmacists’ Interventions on Clinical Significance and Cost Avoidance in a Tertiary Care University Hospital in Oman: A Retrospective Analysis"

_pharmacy, 2022, doi:10.3390/pharmacy10050127_

Round 1
Reviewer 1 Report
Thank you for the opportunity to review this manuscript. This paper examined clinical pharmacists’ interventions in Oman and the clinical and economic benefits from this. The results identified that there was an overall positive clinical and financial impact of clinical pharmacists’ interventions. The results of this study appear to largely be in agreement with, but have extended upon, the existing literature on this topic. The authors should also be commended on conducting this study that fills this gap in the Oman context.
The manuscript is generally well written and discussed, just a few minor errors, but the objectives were clear and methodology sound. Some good points raised in the limitations and future directions, with some important conclusions made and some lessons that can be learnt. Overall, I enjoyed reading this manuscript and learning the findings. I just have a few minor comments.
Specific comments:
· Given that this is an international journal, I think it would be helpful if the country of origin was reflected in the title of the manuscript for the reader. The country of origin from this study was also still unclear from the abstract. I would suggest at least including this information it in the Abstract’s Methods.
· Line 5 Abstract Objectives – Was this supposed to be “…individuals had drawn a great attention…”? If so, please correct.
· Line 44 Introduction – As per last comment, “….drawn attention…”?
· Methods – Can the authors please briefly explain how the actual data analysis/classification allocations were done by the research team as this is not clear – ie. Was the data extracted and analysed and the study measures classified/graded by one pharmacist assessor or several? Any cross checking by more than one pharmacist for consistency for example? I note that this point (potential variability in grading) was briefly alluded to in the limitations in line 259.
· Line 182 Discussion – “….providing accurate dosing information…”. Please correct typo.
· Line 248 Discussion – “…lay the groundwork….”
· Table 1 & 3 – For clarity, I would suggest clarifying that the $ is in the context of USD
· Figure 1 – Please check the numbers in this flow chart. According to “total screened” and “excluded”, the Included for data collection should be n= 2023. Please correct this and the subsequent numbers in this flow chart. As a result, there is also some mismatch with the results in the text eg. line 157 stated n=1824 were analysed for cost impact whereas figure 1 stated n=1969. Please double check the all the n numbers and correct.
· Figure 3 – Graph legend “Avoided unnecessary exposure” (typo “unnecessary”)
Author Response
Reviewer 1
Comments and Suggestions for Authors
Thank you for the opportunity to review this manuscript. This paper examined clinical pharmacists’ interventions in Oman and the clinical and economic benefits from this. The results identified that there was an overall positive clinical and financial impact of clinical pharmacists’ interventions. The results of this study appear to largely be in agreement with, but have extended upon, the existing literature on this topic. The authors should also be commended on conducting this study that fills this gap in the Oman context.
The manuscript is generally well written and discussed, just a few minor errors, but the objectives were clear and methodology sound. Some good points raised in the limitations and future directions, with some important conclusions made and some lessons that can be learnt. Overall, I enjoyed reading this manuscript and learning the findings. I just have a few minor comments.
Our response: Thank you for your supportive summary of our study.
Specific comments:
- Given that this is an international journal, I think it would be helpful if the country of origin was reflected in the title of the manuscript for the reader. The country of origin from this study was also still unclear from the abstract. I would suggest at least including this information it in the Abstract’s Methods.
Our response: We have added in Oman to the title and abstract
- Line 5 Abstract Objectives – Was this supposed to be “…individuals had drawn a great attention…”? If so, please correct.
Our response: Thank you for typo correction, drawn was adjusted
- Line 44 Introduction – As per last comment, “….drawn attention…”?
Our response: Thank you for typo correction, drawn was adjusted
- Methods – Can the authors please briefly explain how the actual data analysis/classification allocations were done by the research team as this is not clear – ie. Was the data extracted and analysed and the study measures classified/graded by one pharmacist assessor or several? Any cross checking by more than one pharmacist for consistency for example? I note that this point (potential variability in grading) was briefly alluded to in the limitations in line 259.
Our response: The study measures were originally classified as per the pharmacy department protocol which is annually being validated by panel of senior pharmacists. The retrieved data were further reviewed in term of cost classification only, without changing other originally selected measures by the clinical pharmacists. For that we have stated in the limitation that “potential variability in grading” due to lacking peer review on all measures.
We have made it more clear in the methods section.
‘We retrieved interventions that were originally documented and classified into several measures by 14 clinical pharmacists over the study period. The captured information included the following: the admitting specialty, prescriber’s’ designation, types and outcomes of the interventions, clinical significance, grading of the clinical significance, and the cost implication associated with each intervention. Forms with incomplete or missing information were excluded. The following study measures were analyzed, however, only cost implication was crossed checked by another clinical pharmacists. ‘
- Line 182 Discussion – “….providing accurate dosing information…”. Please correct typo.
Our response: doing was correct to dosing
- Line 248 Discussion – “…lay the groundwork….”
Our response: ground was correct to groundwork
- Table 1 & 3 – For clarity, I would suggest clarifying that the $ is in the context of USD
Our response: we have adjusted US$ to USD throughout the manuscript as per suggestion.
- Figure 1 – Please check the numbers in this flow chart. According to “total screened” and “excluded”, the Included for data collection should be n= 2023. Please correct this and the subsequent numbers in this flow chart. As a result, there is also some mismatch with the results in the text eg. line 157 stated n=1824 were analysed for cost impact whereas figure 1 stated n=1969. Please double check the all the n numbers and correct.
Our response: Thank you for the important note, figure number was adjusted to 1824.
- Figure 3 – Graph legend “Avoided unnecessary exposure” (typo “unnecessary”)
Our response: Thank you for typo correction, unnecessery was adjusted to unnecessary
Reviewer 2 Report
This research addresses the main question of clinical pharmacists' clinical significance and impacts on cost avoidance. I believe this topic is relevant in the field of clinical pharmacy. I also think that this manuscript adds to the subject area as pharmacoeconomic studies are sparse. The study methodology and references seem appropriate, and the conclusions are consistent with the results.
This is an important study interesting to a broad readership as it shows the impact community pharmacist have on appropriate drug use. I only have a few suggestions:
Please improve the English language through the manuscript
Add article type to the template
Rewrite the abstract to make it more informative and attractive to readers
Figure 1 should be enhanced (text is not showing in N=63)
Author Response
Reviewer 2
Comments and Suggestions for Authors
This research addresses the main question of clinical pharmacists' clinical significance and impacts on cost avoidance. I believe this topic is relevant in the field of clinical pharmacy. I also think that this manuscript adds to the subject area as pharmacoeconomic studies are sparse. The study methodology and references seem appropriate, and the conclusions are consistent with the results.
This is an important study interesting to a broad readership as it shows the impact community pharmacist have on appropriate drug use. I only have a few suggestions:
Our response: Thank you for your supportive comment
Please improve the English language through the manuscript
Our response: we have improved the English as per suggestion.
Add article type to the template
Our response: we have added article type to the title and abstract as per suggestion.
Rewrite the abstract to make it more informative and attractive to readers
Our response: we have rewritten it as per suggestion.
Figure 1 should be enhanced (text is not showing in N=63)
Our response: we have enhanced it as per suggestion, text is now showing.
Reviewer 3 Report
The manuscript evaluates the clinical significance and analysis of the presumed cost avoidance achieved by clinical pharmacists’ interventions.
Results showed that there was an overall positive clinical and financial impact of clinical pharmacists’ interventions.
I find the topic interesting and being worth of investigation and the document is well strucutred, organized, fluidly written, the background is adequate, the methodology is clearly explained and results support the conclusions.
Although I propose the following comments/suggestions:
- keywords should be in alphabetical order
Author Response
Reviewer 3
Comments and Suggestions for Authors
The manuscript evaluates the clinical significance and analysis of the presumed cost avoidance achieved by clinical pharmacists’ interventions.
Results showed that there was an overall positive clinical and financial impact of clinical pharmacists’ interventions.
I find the topic interesting and being worth of investigation and the document is well strucutred, organized, fluidly written, the background is adequate, the methodology is clearly explained and results support the conclusions.
Our response: Thank you for your supportive comment.
Although I propose the following comments/suggestions:
- keywords should be in alphabetical order
Our response: Thank you re-ordered the keywords to be in alphabetical order
Keywords: Clinical Pharmacists; Clinical Significance; Cost Savings; Health Care Costs; Interventions; Oman